# Abundance and diversity of host-seeking adult female mosquitoes in a coastal ecosystem in southern Mexico

Julio César Canales-Delgadillo[1,2]*, Nallely Vázquez-Pérez[3], Vicente Viveros-Santos[4], Rosela Pérez-Ceballos[1,2], José Gilberto Cardoso-Mohedano[2], Arturo Zaldívar-Jiménez[5], Omar Celis-Hernández[1,2], Alejandro Gómez-Ponce[2], Martín Merino-Ibarra[6]

1 Investigadoras e Investigadores por México, Secretaría de Ciencia, Humanidades Tecnología e Innovación (SECIHTI), CDMX, México, 2 Instituto de Ciencias del Mar y Limnología UNAM, Ciudad del Carmen, Campeche, México, 3 Centro de Estudios Tecnológicos del Mar No. 29, Ciudad del Carmen, Campeche, México, 4 Centro de Investigaciones Regionales Dr. Hideyo Noguchi, Universidad Autónoma de Yucatán, Mérida, Yucatán, México, 5 Asesoría Técnica y Estudios Costeros (ATEC), Mérida, Yucatán, México, 6 Ecología y Biodiversidad Acuática, Instituto de Ciencias del Mar y Limnología UNAM, CDMX, México

* jccanalesde@secihti.mx

## Abstract

Mosquito diversity influences disease risk because only certain species transmit pathogens, making the identification of species assemblages essential. To better understand mosquito diversity in the southern Gulf of Mexico, we conducted a study on Isla del Carmen, Campeche, from September 2019 to December 2020. Adult mosquitoes were collected using buccal aspirators during 24-hour cycles in mangrove and low-semideciduous forest patches across three climate seasons: norte, rainy and dry. Sampling occurred every four hours, and species were identified. Hill numbers of order $q=0$, $q=1$, and $q=2$, non-binomial GLMs, NMDS, PERMANOVA, and generalized estimating equations were used to analyze mosquito diversity, abundance, and phenology. We collected 21,424 mosquitoes from 11 genera, 26 species, and four morphospecies. The mosquito abundance and richness peaked during the norte season ($β=1.057$, $z=2.480$, $p=0.013$), with the season being the primary determinant of abundance (PERMANOVA, $F=7.229$, $R^2=0.512$, $p=0.003$). The vegetation type and sampling hour showed effects only when excluding the eudominant *Aedes taeniorhynchus*. The top five genera, *Aedes*, *Psorophora*, *Mansonia*, *Culex* and *Anopheles*, exhibited distinct phenological patterns, with abundance peaking between September 2019 and February 2020. Isla del Carmen is a key region for mosquito diversity in the Yucatan Peninsula, hosting species known to transmit pathogens to humans and wildlife. Our findings highlight the norte season, when cooler temperatures and moderate rainfall are present, as a critical period for mosquito activity, emphasizing the need for targeted vector surveillance and control efforts during this time in the region. This study provides valuable insights into mosquito community dynamics and their implications for public health in coastal areas of southern Mexico.

**Data availability statement:** Data on seasonal species richness and shapefiles for mapping the stuy area are available at https://doi.org/10.5281/zenodo.14633234.

**Funding:** The author(s) received no specific funding for this work.

**Competing interests:** The authors have declared that no competing interests exist.

## Author summary

Mosquitoes are vectors of diseases that affect humans and animals. Despite their global significance, there is still much to learn about their diversity and behavior. We examined host-seeking adult female mosquito diversity, abundance, and activity patterns in southern Mexico's semi-conserved coastal area of Isla del Carmen. Human landing collection sampling was conducted monthly in mangrove and semideciduous forest habitats for over a year, covering an entire 24-hour cycle during different seasons. Among the 26 recorded species, 84% are known vectors of diseases such as dengue, Zika, and encephalitis. The results revealed mosquito diversity and activity peak during the norte season, characterized by cooler temperatures and moderate rainfall. The black salt marsh mosquito, *Aedes taeniorhynchus*, was the most dominant species due to its ability to thrive in the region's brackish and freshwater habitats. Other medically important mosquito species were recorded, highlighting the potential health risks for humans and wildlife. This research provides essential baseline data for understanding mosquito populations in tropical coastal ecosystems, designing disease control strategies, and preserving ecological balance, particularly as climate and land use changes may alter mosquito distribution and activity in the future. This work bridges the gap between mosquito ecology and its public health and conservation implications.

## Introduction

The southern Gulf of Mexico is renowned for its ecological diversity. It encompasses wetlands, jungles, coastal areas, mangroves, and estuaries, providing critical habitats for native and migratory wildlife, including insects [1–3]. Mosquitoes are among the most significant insect vectors of zoonotic diseases, transmitting pathogens affecting human and wildlife health [4–10]. Globally, 3,570 mosquito species have been identified, with approximately 6% occurring in Mexico [11]. These insects are pivotal in transmitting diseases such as malaria, dengue fever, Chikungunya, and Zika virus, which collectively impact millions of people annually, particularly in tropical regions [12–14]. Understanding mosquito diversity and abundance in these areas is essential for developing effective vector-borne disease control strategies [15].

Southern Mexico, including the Yucatan Peninsula, is a hotspot for mosquito-borne diseases due to its tropical climate, high humidity, and abundant water bodies, all of which create ideal conditions for mosquito breeding [16,17]. While some mosquito species, such as *Aedes aegypti* and *Anopheles albimanus*, are well-studied vectors in this region, many other mosquito species' diversity and ecological roles remain poorly understood [18]. Filling these knowledge gaps is critical for enhancing our understanding of mosquito ecology and improving disease prevention efforts in this tropical region.

Isla del Carmen, located off the coast of Campeche, is characterized by its mangrove ecosystems and humid tropical climate, providing numerous microhabitats for mosquito populations [19–21]. Despite its ecological significance, there is limited information about the diversity of mosquito species on the island. Most studies have focused on mainland Campeche, examining mosquito species in low semi-evergreen forests, mangroves, and agricultural landscapes [22,23]. The only known comprehensive survey of mosquito species on Isla del Carmen was conducted between 1996 and 2006, reporting 14 medically relevant species for urban areas [24]. Land use plays a crucial role in shaping mosquito distribution. Thus, the transformation of natural ecosystems can increase the availability of breeding sites and alter species interactions, as habitat degradation and urbanization often create conditions that favor vector species [25].

To address these gaps, we aimed to determine the diversity of host-seeking female mosquito species on Isla del Carmen. We hypothesized that mosquito species richness and diversity are influenced by seasonality, time of day, and vegetation type, which in turn shape the dynamics of disease risk in the region. Our study provides baseline data to inform future research on mosquito populations and the dynamics of vector-borne diseases under changing climatic and ecological conditions. This knowledge is crucial for designing effective disease control and prevention strategies tailored to the unique environmental characteristics of coastal areas in southern Mexico.

## Results

### Species richness and abundance

During the sampling period, we collected 21,424 mosquito individuals, of which 98.60% ($n = 21,124$) were adult females, and 1.40% ($n = 300$) were adult males, divided into two subfamilies (Culicinae and Anophelinae), 11 genera, 26 species, and four morphospecies (Table 1). About 99% of the captured females had an empty abdomen, 147 individuals were completely engorged, and 64 were partially engorged, showing dark blood within the abdomen, meaning that the blood meals were taken less than six hours or between six and eighteen hours before being trapped, respectively. No gravid females were collected during the samplings. The collected males belonged to the species *Ae.* (*Ochlerotatus*) *taeniorhynchus* Wiedemann, 1821 ($n = 263$), *Culex* (*Culex*) *interrogator* Dyar & Knab, 1906 ($n = 15$), *Psorophora* (*Janthinosoma*) *ferox* von Humbolt, 1819 ($n = 10$), *An.* (*Anopheles*) *crucians* Wiedemann, 1828 ($n = 7$), and *Ae.* (*Ochlerotatus*) *angustivittatus* Dyar & Knab, 1907 ($n = 5$). Most of the females were caught on the collectors' bodies. However, some catches (< 1%) were obtained from domestic dogs' bodies (*Ae. taeniorhynchus*, *An. crucians*). Males were caught mainly from buildings' windows and walls (*Ae. taeniorhynchus*, *An. crucians*, *Cx. interrogator*) and from plant branches and leaves (*Ps. ferox*, *Ae. angustivittatus*).

All the genera recorded in our study were present in the norte season, while in the dry season, less than 50% of the recorded genera were present.

The highest species richness was recorded during the norte season ($n = 26$), while it was lower in the rainy and dry seasons ($n = 20$, and $n = 8$, respectively, Fig 1). The black salt marsh mosquito (*Ae. taeniorhynchus*) was the most abundant and dominant species at any time during the 24-hour cycle, sampling season, and vegetation type, representing 92.3% of our sample, which had a masking effect on the ecological response of the remaining species in the community (Table 2).

The genus *Aedes* was the best represented (94.6%), while genera such as *Aedeomyia* Theobald, *Deinocerites*, *Limatus* Theobald, and *Uranotaenia* Lynch Arribálzaga typically had low relative abundances (0.023, 0.284, 0.018, and 0.004%, respectively, Table 1). Seasonally, host-seeking female mosquito abundance was highest during the norte season. Even after excluding the most abundant species (*Ae. taeniorhynchus*), mosquito counts remained higher in this season (Table 3). An increase in female mosquito abundance by hour and vegetation type was observed only when *Ae. taeniorhynchus* was excluded from the analysis. The highest counts occurred at 13:00 and 17:00, and the abundance was also higher in non-modified vegetation types (Table 3).

PLOS Neglected Tropical Diseases

**Table 1. Species and abundance of host-seeking adult female mosquitoes in Isla del Carmen.** Sampling was carried out during a 24-hour cycle (Time) and three sampling seasons: norte (N), dry (D), and rainy (R). The medical and veterinary relevance of mosquito species was determined by the associated pathogens reported in the literature (see references in S1 Appendix). The last three rows show the total abundance by season across all day times.

| Time | 1:00 | 5:00 | 9:00 | 13:00 | 17:00 | 21:00 | Total Abundance | Associated Pathogens |
|---|---|---|---|---|---|---|---|---|
| Season | N/D/R | N/D/R | N/D/R | N/D/R | N/D/R | N/D/R | | |
| Anophelinae | | | | | | | | |
| Anopheles (Anopheles) crucians | 11/0/0 | 4/1/0 | 0/0/0 | 1/0/0 | 5/0/0 | 5/0/0 | **27** | EEEV, TENV, WNV, RVFV, Pf, Di |
| An. (Ano.) gabaldoni | 3/0/0 | 6/0/0 | 0/0/0 | 0/0/0 | 6/0/0 | 2/0/0 | **17** | Unknown |
| An. (Ano.) vestitipennis | 7/0/0 | 5/0/3 | 0/0/0 | 0/0/0 | 1/0/0 | 3/0/0 | **19** | Pv |
| An. (Ano.) sp. | 0/0/0 | 2/0/0 | 0/0/0 | 0/0/0 | 0/0/0 | 2/0/0 | **4** | Not applicable |
| An. (Nyssorhynchus) albimanus | 3/0/1 | 2/0/0 | 0/0/0 | 0/0/0 | 4/0/0 | 0/1/0 | **11** | TENV, Pf, Pv |
| Culicinae | | | | | | | | |
| Aedeomyia (Aedeomyia) squamipennis | 0/0/0 | 0/0/0 | 0/0/0 | 0/0/0 | 0/0/0 | 5/0/0 | **5** | GAMV, VEEV, aP |
| Aedes (Ochlerotatus) angustivittatus | 15/0/1 | 19/0/4 | 55/0/5 | 80/0/13 | 42/0/1 | 29/0/4 | **268** | ILHV, VEEV, Di, *D. hominis* |
| Ae. (Och.) fulvus | 0/0/0 | 0/0/0 | 0/0/0 | 0/0/0 | 0/0/0 | 0/0/2 | **2** | BUNV, MAV, VEEV, UNAV |
| Ae. (Och.) scapularis | 1/0/0 | 3/0/2 | 1/0/5 | 4/0/2 | 3/0/1 | 2/0/3 | **27** | CVV, ILHV, MAYV, KAIV, MAV, MELV, ROCV, SLEV, YFV, WYOV, Wb, Di, |
| Ae. (Och.) serratus | 6/0/5 | 10/0/25 | 5/0/7 | 15/0/18 | 15/0/6 | 4/0/3 | **119** | ILHV, OROV, VEEV, YFV, PIXV, SLEV, TROV, MAV, MELV, VEEV, UNAV, D hominis |
| Ae. (Och.) taeniorhynchus | 1419/1237/730 | 1283/618/648 | 2222/289/885 | 2610/933/900 | 2172/185/546 | 1248/901/676 | **19502** | CVV, EEEV, KAIV, ORIV, RVFV, POTV, TENV, VEEV, WNV, Di |
| Ae. (Stegomyia) aegypti | 15/0/2 | 15/0/3 | 6/0/6 | 12/0/3 | 10/0/0 | 6/0/7 | **85** | CVV, DENV, YFV, ILHV, CHIKV, MAYV, WNV, EEEV, VEEV, WEEV, KRV, RRV, ZIKAV, Di |
| Haemagogus (Haemagogus) regalis | 0/0/0 | 0/0/0 | 0/0/0 | 1/0/0 | 0/0/0 | 0/0/0 | **1** | Unknown |
| Psorophora (Janthinosoma) albipes | 0/0/0 | 17/0/0 | 2/0/1 | 1/0/1 | 0/0/1 | 0/0/0 | **23** | ILHV, IERIV, VEEV, WYOV |
| Ps. (Jan.) ferox | 24/0/4 | 18/0/9 | 51/0/15 | 155/0/76 | 121/0/2 | 24/0/31 | **530** | ARUV, BUNV, IERIV, ILHV, KAIV, UNAV, VEEV, MELV, POTV, ROCV, SLEV, WYOV |
| Ps. (Psorophora) ciliata | 1/0/0 | 1/0/0 | 0/0/1 | 1/0/0 | 0/0/0 | 0/0/1 | **5** | EEEV, VEEV, TENV, WNV, Di |
| Ps. (Pso.) sp. | 0/0/0 | 0/0/1 | 0/0/0 | 0/0/0 | 0/0/0 | 0/0/0 | **1** | Not applicable |
| Culex (Anoedioporpa) restrictor | 9/1/1 | 9/0/0 | 1/0/0 | 0/0/0 | 0/0/0 | 1/1/0 | **23** | Unknown |

*(Continued)*

**Table 1.** (Continued)

| Time / Season | 1:00 N/D/R | 5:00 N/D/R | 9:00 N/D/R | 13:00 N/D/R | 17:00 N/D/R | 21:00 N/D/R | Total Abun-dance | Associated Pathogens |
|---|---|---|---|---|---|---|---|---|
| *Cx. (Culex) coronator* | 0/1/2 | 0/0/0 | 0/0/0 | 0/0/0 | 0/0/0 | 1/0/1 | **5** | CARV, VEEV, SLEV, WNV, ZIKAV |
| *Cx. (Cux.) interrogator* | 3/4/0 | 6/2/14 | 0/0/1 | 2/0/0 | 25/0/0 | 3/0/1 | **61** | CxFV, WNV |
| *Cx. (Cux.) nigripalpus* | 0/0/0 | 0/0/0 | 0/0/0 | 0/0/0 | 1/0/0 | 0/0/1 | **2** | CARV, WNV, SLEV, RVFV, EEEV, VEEV, KEYV, PATV, PIXV, Di |
| *Cx. (Cux.)* sp. | 15/0/0 | 1/0/0 | 0/0/0 | 0/0/0 | 0/0/0 | 3/0/0 | **19** | Not applicable |
| *Cx. (Melanocon-ion) taeniopus* | 0/0/4 | 0/0/6 | 0/0/0 | 0/0/0 | 2/0/0 | 4/0/3 | **19** | EEEV, SLEV, NEPV, VEEV |
| *Cx.* sp. | 6/0/0 | 33/0/0 | 0/0/0 | 0/0/0 | 3/0/0 | 3/0/0 | **45** | Not applicable |
| *Deinocerites pseudes* | 9/28/1 | 3/2/2 | 0/0/1 | 0/0/0 | 4/1/1 | 0/7/1 | **60** | SLEV, VEEV |
| *Coquillettidia (Rhynchotaenia) nigricans* | 2/0/0 | 1/0/0 | 1/0/0 | 0/0/0 | 0/0/0 | 1/0/0 | **5** | Unknown |
| *Cq. (Rhy.) venezuelensis* | 5/0/1 | 1/0/2 | 0/0/0 | 5/0/0 | 2/0/1 | 0/0/0 | **17** | MAYV, OROV, SLEV, WNV |
| *Mansonia (Manso-nia) dyari* | 9/0/13 | 16/0/7 | 0/0/11 | 12/0/36 | 82/0/9 | 11/1/10 | **217** | RVFV |
| *Limatus durhamii* | 0/0/0 | 0/0/0 | 0/0/0 | 0/0/0 | 4/0/0 | 0/0/0 | **4** | VEEV, WYOV, *D. hominis* |
| *Uranotaenia (Uranotaenia) lowii* | 0/0/0 | 0/0/0 | 0/0/0 | 0/0/0 | 0/0/0 | 1/0/0 | **1** | Unknown |
| **Total Norte** | 1563 | 1455 | 2344 | 2899 | 2502 | 1358 | **12121** | |
| **Total Dry** | 1271 | 623 | 289 | 933 | 186 | 911 | **4213** | |
| **Total Rainy** | 765 | 726 | 938 | 1049 | 568 | 744 | **4790** | |

**Viruses:** ARUV (Aruac), BUNV (Bunyawera group), CARV (Carapuru), CHIK (Chikungunya), CVV (Cache Valley), CxFV (Culex flavivirus), DENV (dengue), EEEV (Eastern equine encephalitis), GAMV (Gamboa), IERIV (Ieri), ILHV (Ilheus), KAIV (Kairi), KRV (Kamiti river), MAV (Maguari), MAYV (Mayaro), MELV (Melao), NEPV (Nepuyo), ORIV (Oriboca), OROV (Oropuche), PATV (Patois), PIXV (Pixuna), POTV (Potosi), ROCV (Rocio), RRV (Ross river), RVFV (Rift Valley fever), SLEV (Saint Louis encephalitis), TENV (Tensaw), TROV (Trocara), UNAV (Una), VEEV (Venezuelan equine encephalitis), WNV (West Nile), WYOV (Wyeomyia), YFV (Yellow fever), ZIKAV (Zika). **Protozoa:** Pf (*Plasmodium falciparum*), Pv (*Plasmodium vivax*), aP (aviar Plasmodium). **Nematodes:** Di (*Dirofilaria immitis*), Wb (*Wuchereria brancofti*). Other arthropods: *Dermatobia hominis*.

## Diversity patterns

The diversity analysis revealed values higher than 90% sample coverage in the norte and rainy seasons (94.71 and 92.1%, respectively). A lower sample coverage was estimated for the dry season (81.48%) (see S1 Table). In addition, we had sample coverage higher than 80% during each sampling hour (except at 17:00, where the sample coverage was 75.5%), indicating that our sampling effort was sufficient to determine the host-seeking adult female mosquito species richness in our study area.

The Hill numbers showed that the season with the highest diversity of host-seeking female culicids was the norte season. We observed that common ($q=1$) and dominant species ($q=2$) were significantly different only for the dry season (because no confidence intervals overlapped) [26,27]. According to the Hill numbers, for species richness ($q=0$), the differences between the rainy and norte seasons were not clear, even when we extrapolated our sampling effort twice (Fig 2), meaning that these two seasons had similar diversity patterns. When we considered the diversity of each season

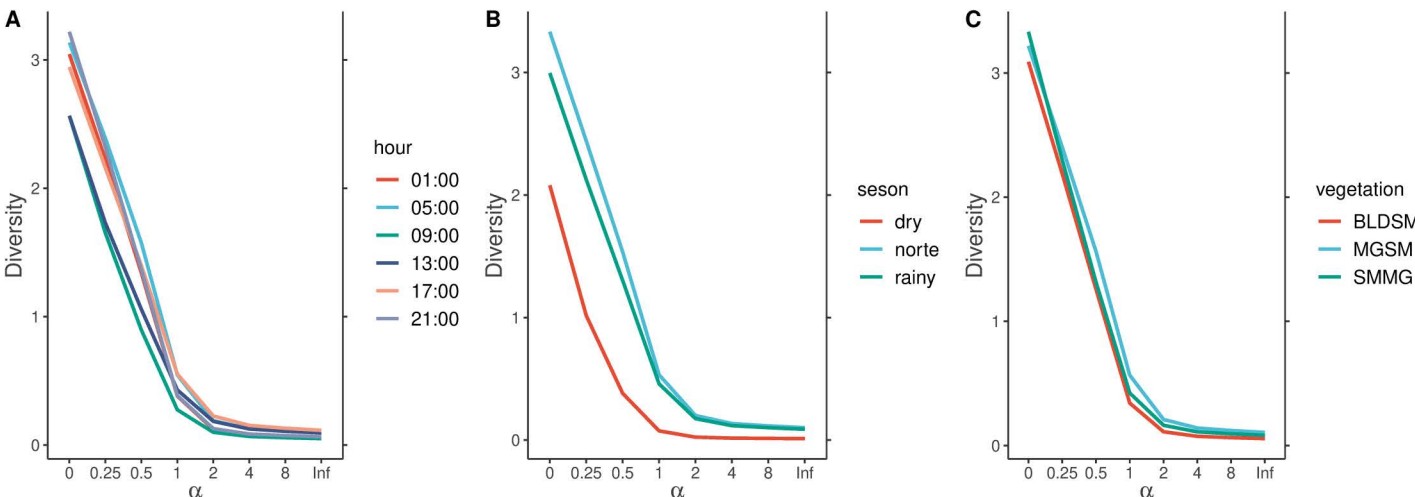

**Fig 1. Rényi diversity profiles of female mosquitoes in Isla del Carmen.** Diversity profiles are shown by time-of-day activity (A), season (B), and vegetation type (C). In the plots, α=0 is the species richness, α=1 is the Shannon−Weiner diversity index, α=2 is the Simpson diversity index, and *Inf* is the Berger−Parker dominance index.

**Table 2. Seasonal and hourly species richness ($S_r$) of the mosquito community in Isla del Carmen.** To show the effect of an eudominant species on the local community, we compared the data including ($H'1$) and excluding ($H'2$) *Ae. taeniorhynchus*.

| | Season | | | Hour | | | | | |
|---|---|---|---|---|---|---|---|---|---|
| | **Dry** | **Rainy** | **Norte** | **01:00** | **05:00** | **09:00** | **13:00** | **17:00** | **21:00** |
| $S_r$ | 8 | 20 | 28 | 21 | 23 | 13 | 13 | 19 | 25 |
| $H'_1$ | 0.075 | 0.459 | 0.533 | 0.369 | 0.488 | 0.274 | 0.429 | 0.553 | 0.301 |
| $H'_2$ | 0.904 | 2.000 | 2.241 | 2.584 | 2.611 | 1.609 | 1.419 | 1.940 | 2.004 |

**Table 3. Estimated seasonal, hourly, and vegetation-type effects on host-seeking female mosquito abundance in Isla del Carmen.**

| | Full dataset | | | | | Filtered dataset | | | | |
|---|---|---|---|---|---|---|---|---|---|---|
| **Season** | **Estimate (β)** | **S. E.** | **z-value** | **p-value** | **exp(β)** | **Estimate (β)** | **S. E.** | **z-value** | **p-value** | **exp(β)** |
| Norte (ref: Dry) | **1.057** | **0.426** | **2.480** | **0.013** | **2.877** | **1.541** | **0.227** | **6.79** | **<0.0001** | **4.669** |
| Rainy (ref: Dry) | 0.128 | 0.426 | 0.301 | 0.763 | 1.137 | **1.399** | **0.247** | **5.67** | **<0.0001** | **4.051** |
| Hour | | | | | | | | | | |
| 05:00 (ref: 01:00) | -0.250 | 0.609 | -0.410 | 0.682 | 0.779 | 0.180 | 0.218 | 0.830 | 0.409 | 1.197 |
| 09:00 (ref: 01:00) | -0.008 | 0.609 | -0.013 | 0.990 | 0.992 | -0.197 | 0.222 | -0.890 | 0.376 | 0.821 |
| 013:00 (ref: 01:00) | 0.305 | 0.609 | 0.500 | 0.617 | 1.356 | **0.721** | **0.214** | **3.370** | **0.001** | **2.056** |
| 017:00 (ref: 01:00) | -0.100 | 0.609 | -0.164 | 0.869 | 0.905 | **0.505** | **0.215** | **2.350** | **0.019** | **1.657** |
| 021:00 (ref: 01:00) | -0.178 | 0.609 | -0.292 | 0.770 | 0.837 | -0.125 | 0.221 | -0.560 | 0.572 | 0.882 |
| Vegetation | | | | | | | | | | |
| SMMG (ref: BLDSM) | 0.096 | 0.207 | 0.465 | 0.642 | 1.101 | **0.478** | **0.155** | **3.080** | **0.002** | **1.613** |
| MGSM (ref: BLDSM) | -0.168 | 0.207 | -0.812 | 0.417 | 0.845 | **0.449** | **0.156** | **2.890** | **0.004** | **1.567** |

The full dataset includes all recorded mosquito species for the negative-binomial GLM estimations. The filtered dataset excluded the eudominant species *Ae. taeniorhynchus* to assess whether its strong dominance influences results for the remaining community species.

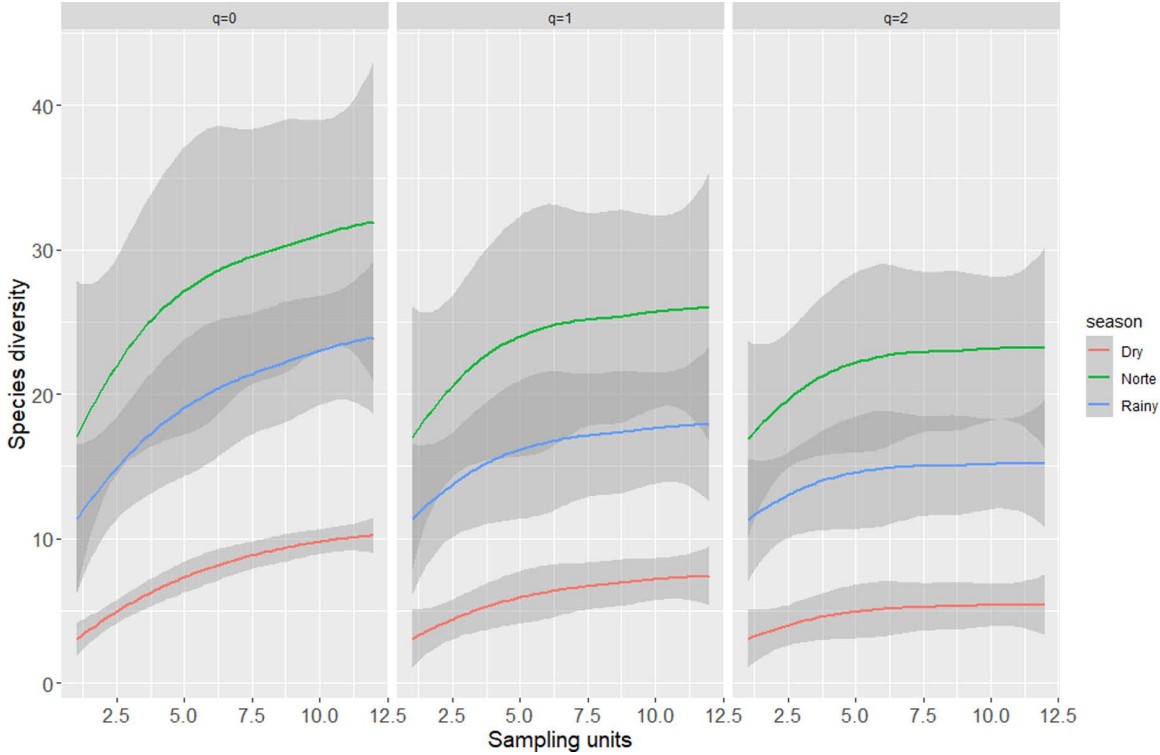

**Fig 2. Estimated Hill numbers for species richness ($q=0$), equally common species ($q=1$), and the effective number of dominant species ($q=2$) during the three sampling seasons.** Confidence interval (shaded areas) overlapping, indicates non-significant differences.

separately, we found that diversity was on average 3.8 times greater in the norte than in the dry season, and that the rainy season was on average 2.6 times more diverse than the dry season.

Rarefaction analysis based on Hill numbers revealed that the species richness differed significantly only between the norte and dry seasons (Fig 3A), indicating a seasonal variation gradient in mosquito community dynamics on Isla del Carmen. Nevertheless, *Ae. taeniorhynchus*, the dominant species, remained highly abundant throughout all seasons, suggesting that annual fluctuations in environmental conditions likely influence the distribution and abundance of the remaining female mosquito species. Although the species richness of host-seeking female mosquitoes appeared to be generally higher between 17:00 and 05:00 hours, no significant differences in species richness, diversity, or dominance were detected across the 24-hour cycle (Fig 3B).

According to the reviewed literature [22,23,28–33], Isla del Carmen is among the places with the greatest species richness on the Yucatan Peninsula, together with Celestún and Calakmul. However, it showed a diversity value similar to that of the urban areas of Merida city [33]. When the eudominant *Ae. taeniorhynchus* was excluded from the dataset, the diversity of the mosquito community in Isla del Carmen was higher than the estimated value for all other locations (Table 4).

## Phenological analysis

NMDS was performed for the whole community in two dimensions with a final stress value of 0.084, indicating a good representation of the data. The ordination revealed clear separation of communities by season (PERMANOVA, $F=7.229$, $R^2=0.512$, $p=0.003$, Fig 4). No significant clustering patterns were found for vegetation types or sampling hour.

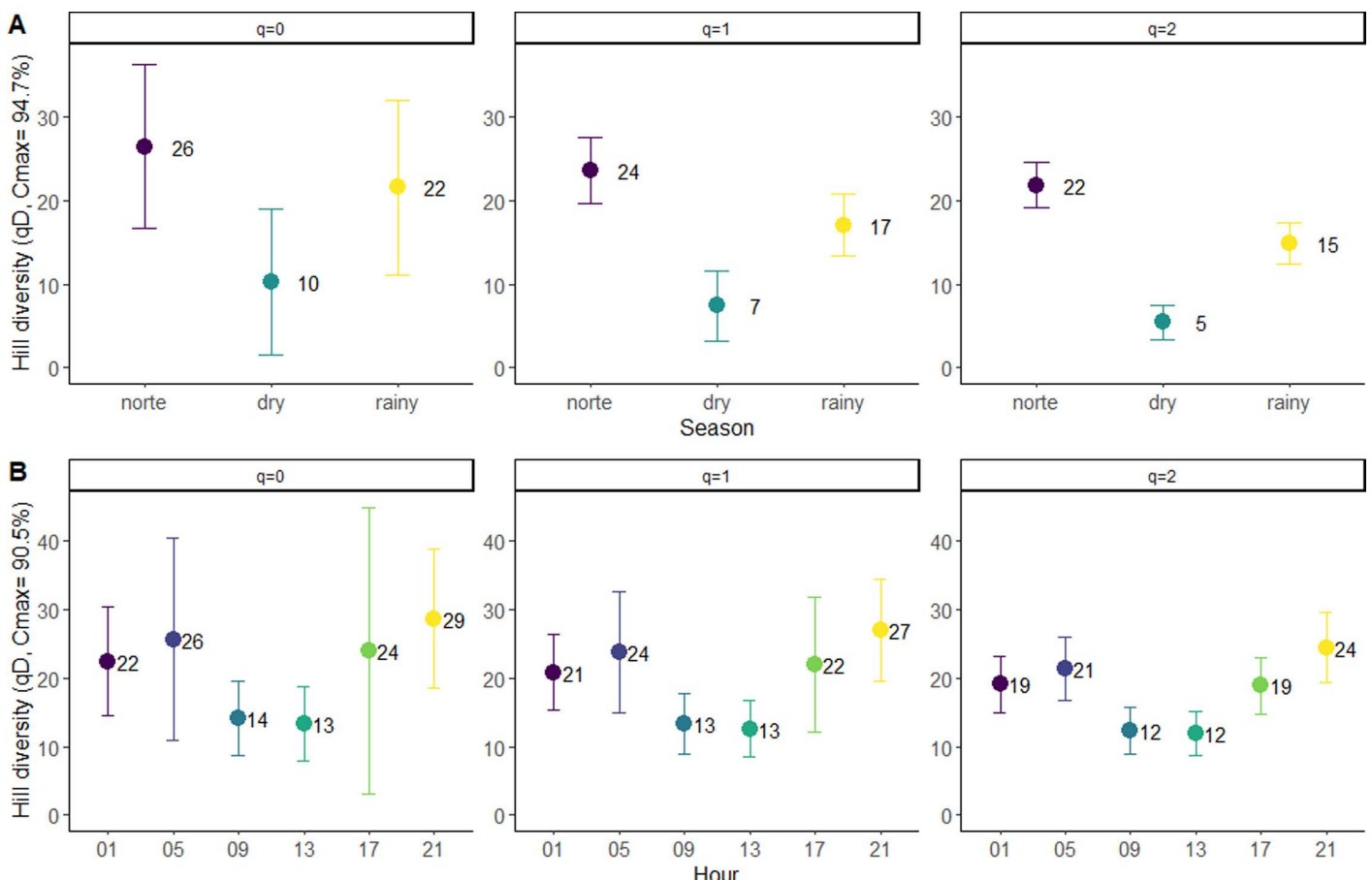

**Fig 3. Expected species richness and diversity patterns.** Rarefied estimates of species richness and diversity patterns in three sampling seasons (A) and six different sampling hours (B). Estimations are based on the sample maximal coverage (Cmax). Vertical bars represent the 95% lower and upper confidence limits.

**Table 4. Comparison of female mosquito species richness ($S_r$) and diversity of Isla del Carmen and other localities in the Yucatan Peninsula. Parameters were estimated from the data in this study and in the earlier published works [23,28,29,33,34].**The estimated Shannon−Wiener diversity index (H') by locality included (H'1) and excluded (H'2) *Ae. taeniorhynchus* data to show the effect of this eudominant species on the community ecological signal. Capture strategy/months are indicated below table with superscripts.

| Parameter | Isla del Carmen[1] | Candelaria[2] | Celestún[3] | Mérida[4] | Palizada[5] | Campeche[6] | Calakmul[7] | Sian kaan[8] |
|---|---|---|---|---|---|---|---|---|
| $S_r$ | 26 | 23 | 33 | 19 | 11 | 6 | 30 | 26 |
| $H'_1$ | 0.447 | 1.607 | 1.629 | 0.751 | 1.377 | 0.752 | 2.089 | 1.761 |
| $H'_2$ | 2.296 | 1.659 | 2.144 | 1.320 | 1.095 | 0.752 | 2.074 | 1.572 |

[1]HLC/Jan-Dec. [2,5]CO$_2$ baited CDC traps/Jul-Aug. [4]DCD backpack aspirator, MMT traps/Aug-Nov. [6]Larval survey/Jan-Dec. [3,7,8]Larval survey, Ovi traps, BG-Sentinel, BDV tent trap/Oct-Nov, Feb-Mar, Jun-Jul, Apr-May.

According to the GEE analysis, the five most abundant mosquito genera exhibited significant phenological differences. *Aedes* species were less frequently captured in mangrove-dominated vegetation compared to areas modified by buildings. Their abundance was higher during the dry season than in the norte season, and their activity was lowest at 21:00. In contrast, *Psorophora* species were more abundant in non-modified vegetation types, showing lower abundance during the

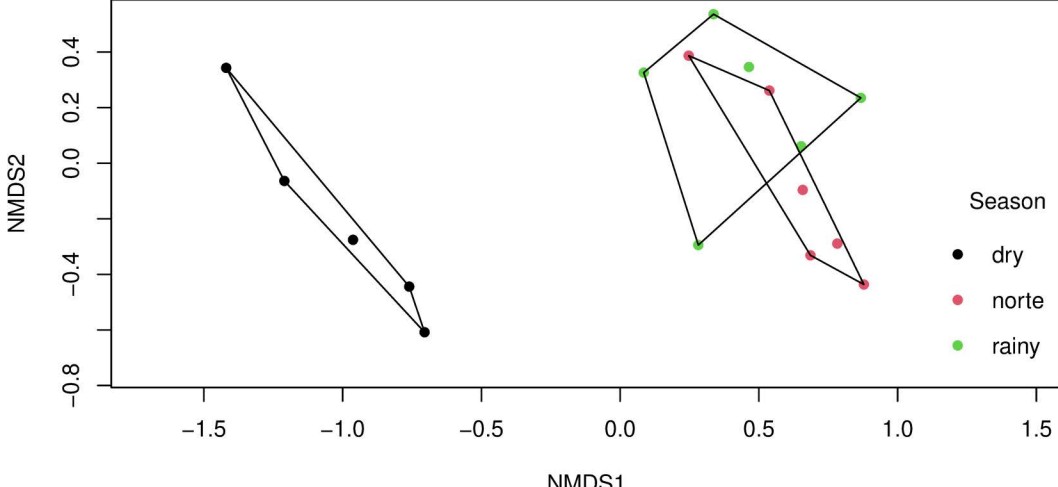

**Fig 4. Non-metric multidimensional scaling (NMDS) ordination of mosquito community.** The data show clear differences in community composition by season. Clustering is based on Bray−Curtis distance, and hulls represent 95% confidence intervals for the seasonal groups.

dry season and exhibiting peak activity between 09:00 and 17:00. *Mansonia* species were predominantly found in semideciduous vegetation, with higher activity around noon and most captures occurring during the norte season (S2 Table).

No significant differences in abundance were detected across vegetation types for *Culex* species. However, they were more abundant during the norte season and exhibited reduced activity from 09:00–13:00. Finally, *Anopheles* species were less abundant in non-modified vegetation compared to modified areas. Their highest abundance was recorded during the norte season, with lower activity observed between 09:00 and 13:00 and at 21:00 (S2 Table).

## Discussion

For the first time, we report the 24-hour cycle dynamics of the Culicidae family during the different climate seasons occurring in Campeche, particularly within the Terminos Lagoon region [35]. By identifying and cataloging the host-seeking adult female mosquito species present in Isla del Carmen, our study provides crucial data on the local and regional mosquito fauna in the southern Gulf of Mexico particularly in the Yucatan Peninsula. This knowledge is foundational for understanding the pathobiology of mosquito-borne diseases, as different mosquito species can carry different pathogens [36,37].

The mosquito population and disease dynamics are affected by environmental factors such as precipitation, temperature, and relative humidity [38], changes in mosquito mortality, and food and breeding site availability [39]. In our study, the norte season had the highest abundance (57.38%) and diversity values, compared to the dry season. This result is similar to that of Romero-Vega et al. [40], who reported that in Costa Rica mosquito diversity is greater in the wet season than in the dry season.

According to Guerra-Santos and Khal [41], the norte and rainy seasons in our study area have very similar monthly rainfall values (range norte season: 19–207 mm, range rainy season: 92–228mm). In addition to rainfall, the norte season typically experiences temperature decreases because of the occurrence of cold fronts, which may favor bionomic conditions [42,43]. During the norte season, the less abundant rainfall helps immature life stages have more favorable breeding site conditions for developing and increasing the number of emerging adults [44], in contrast with overflooding and water flow events that naturally eliminate the immature phases during the rainy season [45]. Roiz et al. [46] reported that rainfall accumulation is positively related to the abundance of several species of *Anopheles*, *Psorophora*, *Uranotaenia*, and subgenera *Ochlerotatus* Lynch Arribálzaga, and *Aedes*, which need large breeding sites at ground level with vegetation and

standing water [47–50]. Such conditions are more likely to occur during the norte season in our study area, increasing the mosquito abundance. Likewise, species such as *An. Albimanus*, *Ae. taeniorhynchus*, *Cx. interrogator*, *De. pseudes* and *Ma. dyari* may breed in a broader range of environmental conditions, since they were present throughout the study period. Furthermore, during the norte season, cooler temperatures allow more mosquito species to experience activity during the day because of lower dehydration and rainfall probabilities. For example, Drakou et al. [39], reported that the optimal temperature for mosquito activity ranges from 15 °C to 24 °C, while above 28 °C, the activity decreases. During the norte season, the average temperature in our study area was about 2.5 °C lower than in the rainy and dry seasons (25.64 °C, 28.32 °C, and 28.23 °C, respectively) [51], which might have favored increased mosquito abundance.

Our study did not find differences in the incidence of host-seeking female mosquitoes at any of the sampling hours or vegetation types. Only when excluding the eudominant *Ae. taeniorhynchus* was there a significant increase in mosquito activity at specific hours, and habitat preferences were detected. It is known that in communities with a highly dominant species, the community-level patterns are influenced by the dominant species' traits, niche overlapping with less abundant species, or by skewing data distribution that can mask responses or ecological signals of the other species in the community [52–54]. By excluding *Ae. taeniorhychus* from the analysis, the other species' responses become apparent, allowing the detection of phenological signals. While the genus *Anopheles* is generally considered nocturnal or crepuscular [55], we observed diurnal activity in several species, including *An. crucians*, *An. gabaldoni*, and *An. vestitipennis*, which were inactive only at sunrise and 09:00. These findings align with reports of medically important species exhibiting similar patterns [56]. In our study, typically diurnal species such as *Ae. angustivittatus*, *Ae. scapularis*, *Ae. serratus*, *Ae. aegypti*, and *Ps. ferox* were observed at 09:00, 13:00 and 17:00, but also at 21:00, 01:00 and 05:00, suggesting behavioral flexibility. Recent studies indicate that artificial light at night may disrupt the mosquitoes natural rhythms, leading to increased nocturnal host-seeking activity, particularly in urban and peri-urban environments [57,58], demonstrating that this shift is not incidental but actively stimulated by artificial light exposure [57–59]. However, further research is needed to investigate similar responses of the species recorded in this study to light pollution. Notably, *Mansonia dyari*, a nocturnal species, was active throughout all sampling hours during the rainy and norte seasons, potentially reflecting seasonal influences on activity patterns [60,61]. The co-occurrence of diurnal and nocturnal species during crepuscular periods highlights the potential for temporal niche overlap, particularly under transitional (natural or artificial) light conditions [42]. However, these patterns may have been influenced by the presence of collectors [43,57], underscoring the need for future studies to employ alternative sampling methods to minimize bias and better understand the full scope of mosquito diversity.

Despite achieving a sampling coverage of 81–94%, our analyses indicated the presence of several undetected species, likely because of the implemented HCL technique. This result suggests that additional sampling effort and complementary collection methods may be necessary to capture rare or elusive species, particularly given the high prevalence of singletons commonly observed in entomological surveys [62].

Isla del Carmen harbors a significant proportion of the mosquito diversity reported for the broader region. Specifically, our findings account for 44% of the species documented in the state of Campeche [23,28,29], 35% of those reported in Yucatan [32,33], and approximately 30% of the species recorded in Quintana Roo [28,29,31]. Furthermore, the species richness observed in our study represents approximately 29% of the mosquito diversity recorded across the Yucatan Peninsula [11]. These differences could be reduced by sampling additional habitat types (marshes, coastal dunes, urbanized areas) In Isla del Carmen, and including collection methods mentioned in Table 4 (for example, $CO_2$-baited traps).

This level of species richness underscores the ecological significance of Isla del Carmen as a regional hotspot for mosquito diversity. The island's unique environmental conditions likely contribute to its role in supporting a wide range of mosquito species, many of which are of medical and veterinary importance (S1 Appendix). Consequently, Isla del Carmen may play a critical role in regional disease dynamics, particularly as a potential reservoir or transmission hub for vector-borne pathogens that affect public health.

## Medical significance of the mosquito species in Isla del Carmen

In total, 84% of the mosquito species recorded in this study are recognized vectors of at least 35 pathogen agents, including arboviruses, flaviviruses, and protozoans, which have significant implications for human, domestic animal, and wildlife health (Table 1 and S1 Appendix). Among these pathogens are West Nile virus, Rift Valley fever virus, Eastern and Western equine encephalitis viruses, *Dirofilaria immitis*, *Dermatobia hominis*, dengue virus, Zika virus, and Chikungunya virus (Table 1 and S1 File).

The species richness in Isla del Carmen was comparable to other regions of the Yucatan Peninsula, but its diversity was lower due to the overwhelming dominance of *Ae. taeniorhynchus*, which accounted for approximately 92% of captured adult females. This eudominant species thrives in brackish water habitats [63], which are abundant in the estuarine and mangrove environments surrounding Isla del Carmen. These conditions favor *Ae. taeniorhynchus* over other species requiring freshwater for breeding, such as *Ae. aegypti* [64] *Culex coronator* [65], *An. crucians*, *An. Vestitipennis*, and *An. gabaldoni* [66]. Additionally, *Ae. taeniorhynchus* is a migratory species capable of traveling 30–96 km [67], enabling it to exploit both brackish and freshwater habitats during rainfall (June–September) and estuarine flooding seasons (November–February) [19,41]. Its abundance poses significant health risks, as it is implicated in the transmission of several pathogens, such as VEEV, EEEV, WNV, and *Dirofilaria immitis*, affecting humans and animals (see Table 1, S1 File, and references in S1 Appendix). While other *Aedes* species were less abundant, they also play roles in pathogen transmission (Table 1, S1 File and S1 Appendix).

Despite being in a semi-conserved area, *Ae. aegypti*, the sixth most abundant species in our study, was present in the study site, likely due to habitat modifications from nearby human settlements (about 450 m apart). This species is highly adaptable, exploiting both artificial water containers [64] and natural breeding sites, such as tree holes during the rainy season [68]. Its activity throughout all sampling hours, influenced by prey presence and artificial light [57,58], underscores its potential as a vector of regional significant diseases such as dengue and Zika.

We recorded three of the four *Psorophora* species previously reported in the Yucatan Peninsula, all of which are potential vectors for pathogens such as Ilheus virus (ILHV) and Venezuelan equine encephalitis (VEE) [36,69]. The presence of migrant bird species flying from North, Central, and South America [70], which can act as reservoirs, increases the risk of zoonotic transmission to humans and wildlife [71–73]. Additionally, we documented *Haemagogus regalis*, which, while not directly implicated in pathogen transmission, deserves monitoring due to the presence of wild primate populations and the potential for wild-type yellow fever virus (YFV) spillover [74], since other *Haemagogus* species are involved in the transmission of sylvatic YFV type [75].

Five *Culex* species were identified in Isla del Carmen, with two remaining unidentified due to morphological limitations. Known vectors of West Nile virus (WNV), St. Louis Encephalitis (SLEV), Zika virus (ZIK), and VEE [76–78], these species pose significant health risks, particularly in rural and estuarine areas [79,80]. Similarly, two *Coquillettidia* species were recorded, with *Cq. venezuelensis* identified as a vector for multiple pathogens affecting humans and wildlife [72,81] (S1 Appendix, S1 File).

Only *Mansonia dyari* was found among the two *Mansonia* species reported in the Yucatan Peninsula. While its medical relevance is uncertain, it has been implicated in Rift Valley fever virus (RVF) transmission in other regions [82]. Given the presence of migrant birds and livestock in adjacent areas, further investigation is needed. Among the four *Anopheles* species recorded, *An. albimanus* and *An. vestitipennis* are primary malaria vectors in southern Mexico [83,84], while *An. crucians* transmits *Dirofilaria immitis*, which infects domestic animals and rarely humans [85,86].

Although this study was conducted over one instead of two years, as most seasonal pattern studies, it represents the first research in this area in 17 years and documented about twice the number of mosquito species previously recorded, providing valuable baseline data of ecological and public health concerns.

## Conclusion

This study provides the first comprehensive analysis of the 24-hour activity patterns and seasonal dynamics of mosquito populations in Isla del Carmen, a region within the Terminos Lagoon area of the Yucatan Peninsula. Our findings reveal similar species richness to other southern Gulf of Mexico locations, with *Aedes taeniorhynchus* dominating the mosquito community due to its adaptation to coastal habitats and migratory behavior [67]. The presence of this eudominant species, along with other medically significant vectors such as *Ae. aegypti*, *Culex coronator*, *Anopheles albimanus*, and *Psorophora ferox*, underscores the potential public health risks posed by mosquito-borne diseases in this region. Notably, the apparent adaptability of these species to both natural and human-modified environments highlights the need for targeted surveillance, particularly in areas where human activity intersects with natural habitats.

The seasonal variations in mosquito abundance and diversity, particularly during the norte season, emphasize the influence of environmental factors such as temperature, rainfall, and habitat availability on mosquito population dynamics. The co-occurrence of diurnal and nocturnal species suggests that temporal niche overlap may facilitate pathogen transmission and increase the risk of zoonotic spillover. Furthermore, the presence of a growing human population, migrant bird species and wild primates in the area raises concerns about the potential introduction and spread of pathogens such as dengue virus, YFV, WNV, and VEEV.

Given the ecological significance of Isla del Carmen as a natural protected area and its role as a regional hotspot for mosquito diversity, our findings have critical implications for public health and wildlife conservation. Health authorities should prioritize integrated vector management strategies that consider both human and wildlife health, particularly in the context of habitat modification and climate change, which may further alter mosquito dynamics and disease transmission patterns.

## Materials and methods

### Study area

Isla del Carmen is a sandbar island located southwest of the Mexican state of Campeche, within the Yucatan Peninsula (Fig 5). The climate is warm subhumid with minimal and maximal average temperatures ranging from 22°C to 32°C and approximately 1155 mm of rainfall annually. Around the year, the relative moisture content is approximately 74%. The dominant vegetation types are mangroves and semideciduous forests.

The sampling site (vertexes: 18.6517° N -91.7591° W; 18.6507° N -91.7604° W; 18.6555° N -91.7641° W; 18.6567° N -91.7626° W, Fig 1) is a ten-hectare area, modified by the presence of buildings for academic research and embedded in a changing landscape of growing human settlements located at an approximate average distance of 546 m from the center of the sampling area. Although this human influence is evident, within the sampling area four species of mangrove trees (*Rizophora mangle* L., *Laguncularia racemosa* (L.) C. F. Gaertn, *Avicennia germinans* (L.) L., *Conocarpus erectus* L.), form an ecotone with a vegetation community composed of *Metopium brownei* (Jacq.) Urb., *Bursera simaruba* (L.) Sarg. 1890, *Sabal mexicana* Mart., *Lonchocarpus hondurensis* Benth., and *Cedrela odorata* L., among other plant species typical of tropical semideciduous forests. Therefore, the association between mangroves and the tropical semideciduous forests within our study site represents the typical vegetation community of disturbed and natural habitats in Isla del Carmen. There are three regional climate seasons: nortes or cold front season (referred to as norte season in this article) from October to February; dry season from March to May; and rainy season from June to September [35,41].

### Sampling

To assess the host-seeking female mosquito species diversity, we carried out monthly samplings, from September 2019 to December 2020 (permit number: SGPA/DGVS/0408019). The sampling included three collectors that used pooters (oral aspirators) to trap live-flying adult female mosquitoes on three sampling transects established in 1) a vegetation mixture

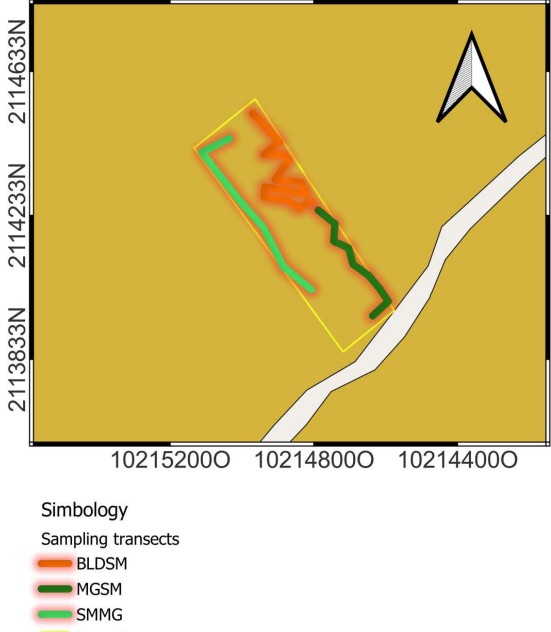

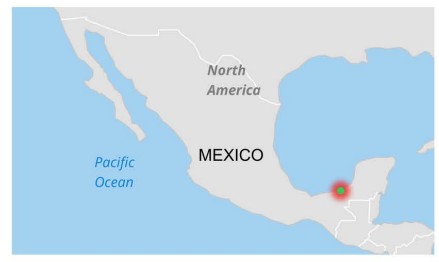

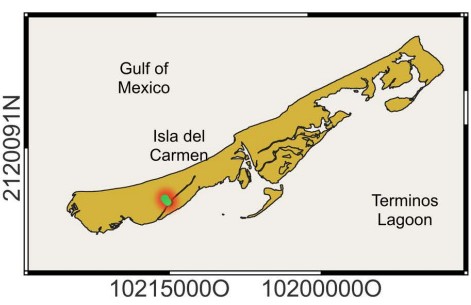

**Fig 5. Study site and sampling transects.** The transects were separated by dominant vegetation: semideciduous forest modified by buildings (BLDSM), mangrove and semideciduous forest (MGSM), and semideciduous and mangrove forest (SMMG). The map was made with Natural Earth. Free vector and raster map data @ naturalearthdata.com, by masking the island shape. Study site and sampling transects were drawn from GPS data points (shapefiles available at https://doi.org/10.5281/zenodo.14633234). Map units in meters.

of dominant mangrove and semideciduous forest (MGSM), 2) a mixture of dominant semideciduous and mangrove forest (SMMG), or 3) a semideciduous forest modified by buildings (BLDSM, Fig 5). Individuals' collection was performed mainly through the human landing collection (HLC) method [87]. The HLC technique is one of the oldest methods to catch mosquitoes and has been widely used for its simplicity in collecting host-seeking adult females [87,88]. However, some adult mosquitoes were collected from non-human organisms, including dogs, buildings' windows and walls, and plant branches and leaves.

According to the Regulations of the General Health Law on Health Research [89], consent for HLC is not required in Mexico. However, all collectors were fully informed about the potential infection risks, and all necessary protective measures were implemented to ensure their well-being and safety during the collection process (use of denim clothes and latex gloves to protect as much skin as possible, IRB approval CEID-SOL-2025). Once the collectors agreed with the working conditions, the sampling began. During the study period, the collectors did not show any infection symptoms.

During the sampling, we covered 16 24-hour cycles consisting of six 60 min trapping sessions per cycle separated by four hours each: in the mornings at 05:00 and 09:00, in the afternoons at 13:00 and 17:00, and in the night at 21:00 and 01:00 [42]. In addition, for each sampling session, one collector walked independently at its pass along one of the three sampling transects (Fig 5) for 60 min to trap mosquito females seeking a host. No speed control was imposed, but each transect had to be entirely walked. Each collector was equipped only with an LED head lamp for secure walking after sunset, one 1 L plastic container for mosquito storage, and a pooter composed of a 30 cm glass tube connected to an 85 cm latex hosepipe with filters at each extreme of the pipe. No UV light or other attractors were used for sampling. Thus, the sampling effort was six hours per transect which summed 18 hours per sampling, for a total of 288 hours of sampling time across the three climate seasons in our study area. The trapped individuals were stored in 1 L plastic containers until they

were transported to the laboratory for sacrifice via thermal shock (-20°C) for 5–10 mins. All the trapped individuals were inspected under a stereoscope (Stemi 305, Carl Zeiss Oberkochen, Germany) for taxonomical identification using specialized keys such as Carpenter and LaCasse [90], Díaz-Nájera [91], Arnell [47], Clark-Gil and Darsie [92], and Wilkerson et al. [93]. Once separated by species, all the mosquitoes were counted, and some specimens were prepared and mounted for a reference collection, which was deposited at the Biodiversity and Conservation Genetics laboratory at the Institute of Sea Sciences and Limnology, El Carmen, UNAM.

### Data analysis

For diversity analyses, the data were grouped according to season (dry, rainy, and norte) of each collection event and the 24-hour sampling cycle. Species richness was estimated using the function *specnumber* of the R package vegan [94].

To evaluate the sampling coverage during different seasons and at different times of the 24-hour cycle, we generated accumulation diversity curves. After the conversion of the abundance data to incidence data, we generated three estimates of Hill numbers of order $q$: 1) species richness ($q = 0$); 2) Shannon diversity, as the exponential of Shannon entropy or common species ($q = 1$); and 3) Simpson diversity, calculated as the inverse of the Simpson concentration or dominant species ($q = 2$), using the functions *iNEXT* and *estimateD* in the R package iNEXT, [26,27]. To visualize the diversity patterns of the mosquito community, we calculated the Rényi diversity profiles (α), which provide a comprehensive view of species diversity by incorporating multiple diversity indices into a single framework through the function *renyicomp* of the BiodivesityR package [95]. When α = 0, the Rényi diversity corresponds to species richness; when α = 1, it approximates the Shannon index; and when α = 2, it is equivalent to the Simpson index, which emphasizes species dominance. As α increases, the diversity becomes more influenced by the most abundant species, allowing for a comparison of evenness across communities. We generated Rényi diversity profiles for each sampling season, hour, and vegetation type to compare mosquito assemblages. The profiles were interpreted by assessing how diversity changed across different α values, with curves that remained above others indicating consistently higher diversity across all scales.

To further analyze the effects of season (norte, dry, and rainy) and sampling hour (01:00, 05:00, 09:00, 13:00, 17:00, and 21:00) on the abundance of host-seeking female mosquitoes, we used a Negative Binomial Generalized Linear Model (GLM) implemented with the *glm.nb* function from the MASS package [96] in R. This approach was chosen due to the presence of overdispersion in the count data, which violates the assumptions of a Poisson GLM. The Negative Binomial GLM accounts for overdispersion by introducing a dispersion parameter, making it suitable for modeling mosquito abundance data [97]. We applied a Non-Metric Multidimensional Scaling (NMDS) analysis to test clustering patterns across seasons and time of day. NMDS was used to visually explore and reinforce the results of the Negative Binomial GLM, providing a complementary perspective on mosquito activity trends. This ordination approach allows identification of seasonal and diel peaks of organisms' activity [98], particularly important for areas where species involved in transmitting diseases such as dengue, Zika, and Chikungunya are present. The NMDS was based on a Bray−Curtis dissimilarity matrix derived from species abundance data. The significance of the clustering patterns was tested through a PERMANOVA test using the function *adonis2* from the R package vegan [94]. Additionally, to assess the effects of season, time of day, and vegetation type on the abundance of the five most dominant mosquito genera in the study area, we employed a Generalized Estimating Equations (GEE) approach using the R package *geepack* [99] to analyze phenological patterns. The GEE model was selected because it accounts for correlated data (e.g., repeated measures across time and space) and allows for robust inference in the presence of non-independence [100,101]. Unlike traditional parametric tests, GEE does not require strict assumptions about the distribution of the dependent variable, making it a suitable choice for our dataset. Only genera with more than 60 records over the study period were included in the analysis to ensure reliable estimates. All analyses were performed in R version 4.1.1 [34].

To better comprehend the contribution of the Isla del Carmen mosquito species to regional diversity, we compared our data with diversity reports in other areas of the Yucatan Peninsula. We compiled the published works with available abundance data [23,28–33], to estimate species diversity and richness in each state/location as described above.

## Supporting information

**S1 Table. Sample coverage estimates derived from host-seeking female mosquito incidence data across different sampling seasons and hours.** The table shows key metrics including sample size (T), total number of incidences (U), observed species richness (S.obs), estimated sample coverage (SC), and the first ten incidence frequency counts (Q1-Q10). Data are categorized by sampling season (Norte, Dry, Rainy) and sampling hour (01:00, 05:00, 09:00, 13:00, 17:00, 21:00).
(DOCX)

**S2 Table. Estimated effects of environmental covariates on the abundance of host-seeking female mosquitoes for the top five most abundant genera in the study area.** The table presents the estimated effects of vegetation type, season, and sampling hour on mosquito abundance for the genera *Aedes*, *Psorophora*, *Mansonia*, *Culex*, and *Anopheles*. Vegetation types include semideciduous forest modified by buildings (BLDSM), dominant semideciduous forest & mangrove (SMMG), and dominant mangrove & semideciduous forest (MGSM). Results are shown as estimates with standard errors, Wald χ² statistics, and corresponding *p*-values.
(DOCX)

**S1 Appendix. List of mosquito species involved in pathogen transmission and their associated references.** Each entry includes the mosquito species, the pathogen it transmits, and the corresponding reference with its DOI or URL for further reading.
(PDF)

**S1 File. Interactive Sankey diagram showing the potential pathogens in the study area, their associated mosquito species, and the hosts they may affect.** This diagram provides a visual representation of the relationships between pathogens, mosquito vectors, and host species, highlighting potential transmission pathways in the study area.
(HTML)

## Acknowledgments

We thank Lourdes Potenciano, Hernán Álvarez and Andrés Reda from ICML-UNAM EL Carmen for supporting the logistics of field work and the sampling sessions.

## Author contributions

**Conceptualization:** Julio César Canales-Delgadillo, Nallely Vázquez-Pérez.

**Data curation:** Vicente Viveros-Santos.

**Formal analysis:** Julio César Canales-Delgadillo, Vicente Viveros-Santos.

**Investigation:** Julio César Canales-Delgadillo, Nallely Vázquez-Pérez, Rosela Pérez-Ceballos, José Gilberto Cardoso-Mohedano.

**Methodology:** Julio César Canales-Delgadillo, Nallely Vázquez-Pérez, Vicente Viveros-Santos, Arturo Zaldívar-Jiménez.

**Supervision:** Julio César Canales-Delgadillo, Rosela Pérez-Ceballos, Martín Merino-Ibarra.

**Validation:** Julio César Canales-Delgadillo, Vicente Viveros-Santos, José Gilberto Cardoso-Mohedano.

**Visualization:** Omar Celis-Hernández, Alejandro Gómez-Ponce, Martín Merino-Ibarra.

**Writing – original draft:** Julio César Canales-Delgadillo, Nallely Vázquez-Pérez, Vicente Viveros-Santos.

**Writing – review & editing:** Julio César Canales-Delgadillo, Rosela Pérez-Ceballos, José Gilberto Cardoso-Mohedano, Arturo Zaldívar-Jiménez, Omar Celis-Hernández, Alejandro Gómez-Ponce, Martín Merino-Ibarra.

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
