## [Decision Letter · Decision Letter 0]

PNTD-D-24-00875

Seasonal and hourly diversity patterns of anthropophagous female mosquito species in a semi-conserved area at the southern Mexico.

Dear Dr. Canales-Delgadillo,

Thank you for submitting your manuscript to PLOS Neglected Tropical Diseases. After careful consideration, we feel that it has merit but does not fully meet PLOS Neglected Tropical Diseases's publication criteria as it currently stands. Therefore, we invite you to submit a revised version of the manuscript that addresses the points raised during the review process.

Please submit your revised manuscript within 60 days due on 11th February 2025. If you will need more time than this to complete your revisions, please reply to this message or contact the journal office at plosntds@plos.org. Please include the following items when submitting your revised manuscript:

We look forward to receiving your revised manuscript.

Kind regards,

Gissella Vasquez

Guest Editor

Amy Morrison

Section Editor

Shaden Kamhawi

co-Editor-in-Chief

Paul Brindley

co-Editor-in-Chief

**Additional Editor Comments:**

The manuscript has interesting and unique information on Yucatan peninsula mosquito diversity and is potentially acceptable for publication, however, all three reviewers had several comments and suggestions to improve its quality. Two of them recommended a major revision and one a minor one. Overall, reviewers suggested to revise the introduction for clarity, improve methods section by providing more details, address whether IRB approval was obtained for human landing collections, perform GLM instead of non-parametrical tests, improve clarity of results and discussion sections considering that sampling did not only target anthropophagous mosquitoes among other relevant recommendations. Two reviewers expressed editorial concerns that include considerable editing and to re-analyze the data. I agree with reviewer's comments and would like to request a major revision. This revision shall address all the issues raised by reviewers. Regarding data analysis, I recommend performing GLM and transforming the data, but please provide a justification or alternative if not.

**Journal Requirements:**

1) Please provide an Author Summary. This should appear in your manuscript between the Abstract (if applicable) and the Introduction, and should be 150-200 words long. The aim should be to make your findings accessible to a wide audience that includes both scientists and non-scientists. Sample summaries can be found on our website under Submission Guidelines:

3) Some material included in your submission may be copyrighted. According to PLOSu2019s copyright policy, authors who use figures or other material (e.g., graphics, clipart, maps) from another author or copyright holder must demonstrate or obtain permission to publish this material under the Creative Commons Attribution 4.0 International (CC BY 4.0) License used by PLOS journals. Please closely review the details of PLOSu2019s copyright requirements here: PLOS Licenses and Copyright. If you need to request permissions from a copyright holder, you may use PLOS's Copyright Content Permission form.

Potential Copyright Issues:

- Figure 1. Please provide a direct link to the base layer of the map (i.e., the country or region border shape) and ensure this is also included in the figure legend; and provide a link to the terms of use / license information for the base layer image or shapefile. We cannot publish proprietary or copyrighted maps (e.g. Google Maps, Mapquest) and the terms of use for your map base layer must be compatible with our CC BY 4.0 license.

4) We note that your Data Availability Statement is currently as follows: "Data on seasonal species richness and abundance are available by request to the correspondence author.". Please confirm at this time whether or not your submission contains all raw data required to replicate the results of your study. Authors must share the “minimal data set” for their submission. PLOS defines the minimal data set to consist of the data required to replicate all study findings reported in the article, as well as related metadata and methods (https://journals.plos.org/plosone/s/data-availability#loc-minimal-data-set-definition).

- The points extracted from images for analysis..

**Reviewers' Comments:**

Reviewer's Responses to Questions

**Key Review Criteria Required for Acceptance?**

**Methods**

-Are the objectives of the study clearly articulated with a clear testable hypothesis stated?

-Is the study design appropriate to address the stated objectives?

-Is the population clearly described and appropriate for the hypothesis being tested?

-Is the sample size sufficient to ensure adequate power to address the hypothesis being tested?

-Were correct statistical analysis used to support conclusions?

-Are there concerns about ethical or regulatory requirements being met?

Reviewer #1: Thanks for having me as a reviewer.

First at all, the Introduction lay out is far from being publishable as it reads quite disorganized and with a lack of flow. The onset talks about topics that are very disparate and not related to what the paper is about. I suggest the authors concentrate on the main aims of the paper (diversity and abundance of mosquitoes) and write their introduction accordingly. For example, the first two paragraphs are really distracting.

I have some issues in relation to whether this study fits into de journal scope. To remedy this, authors should emphasize and/or frame their study aims within a zoonotic background, for example mentioning that they investigated mosquitoes of medical importance to inform local health authorities. To be coherent with this, I suggest that authors divide mosquitoes of medical importance vs those that are not in the analysis. This would imply setting new objectives.

You need to stress why adding land use type is important. Your introduction says nothing about it. You may like to read Rodríguez-González, S., Izquierdo-Suzán, M., Rocha-Ortega, M., & Córdoba-Aguilar, A. (2024). Vector mosquito distribution and richness are predicted by socio-economic, and ecological variables. Acta Tropica, 254, 107179.

I am not at all happy with the use of non-parametrical tests. Did you try to transform your variables?

I suggest adding a specific analysis to detect phenological patterns. This information is key to inform health authorities as for when they should be worried in regards to mosquitoes of medical importance. I am adding a in-revision manuscript from my group that may be of use to authors. Please do not disseminate it as it is not published yet.

Reviewer #2: The methods need revision. See comments below.

Reviewer #3: The framing of the objective of this paper could be improved through improved clarity of the introduction, especially regarding the ways in which increased knowledge about mosquito biodiversity can improve risk assessment for vector-borne diseases.

The study design was appropriate, although certain details of the methods require further explanation (line-by-line comments in attached document). For example, were the mosquitoes caught near dogs held separately to see if there were differences in species attracted to dogs and humans? If not, then it isn’t quite accurate to say you were looking just at anthropophagous mosquitoes, but rather host-seeking mosquitoes in general. Similarly, there was no mention of collection on plants / buildings in your methods, but this was mentioned in your results. Also, more details should be provided about how the diversity metrics were calculated (e.g. provide R packages/functions).

The site was clearly described.

Regarding the statistical analysis, I believe that a generalized linear mixed model would be a more appropriate analysis approach than Kruskal-Wallis because it would allow you to account for pseudo-replication by transect site and collection date, and would also allow you to include various parameters of interest in the same model.

I do not see any mention of an IRB or equivalent review, but human landing catches were utilized to collect mosquitoes, which I believe is considered to involve human subjects.

**Results**

-Does the analysis presented match the analysis plan?

-Are the results clearly and completely presented?

-Are the figures (Tables, Images) of sufficient quality for clarity?

Reviewer #1: Partly. I'd suggest that the analysis is divided for mosqutioes of medical importance vs mosquitoes of non-medical importance. I also suggest carrying out an analysis of phenology.

Reviewer #2: The results present some issues that must be addressed. See specific comments below.

Reviewer #3: The results are presented relatively clearly, but specific lines where further clarification is necessary are outlined in the attached document. Some examples:

Table 1: can you please provide a citation for each mosquito- pathogen association?

Why do you include species richness in your title? I would just highlight seasonal and temporal abundance by mosquito species. Also, you wrote “anthropophagous” mosquitoes in the title; does this mean that this does not include any mosquitoes collected on dogs or plants/buildings?

I think it would be helpful to have another column that gives the total by season across all day times.

Figure 2: it is hard to see the 6 hours since the lines are largely overlapping; it might be better to group by day, night and dusk/dawn.

This is the first mention of the three vegetation type groups with acronyms; please define.

I don’t understand what the Renyi diversity profile is or how it was created; can you please provide more information?

247: How was the diversity analysis conducted?

253-254: Significance and non-overlapping of confidence intervals are not synonymous in most analyses; if this is different for Hill numbers, please provide a reference

255: Not clear according to what test?

Figure 4: numbers overlap the points, making it hard to read.

Table 3: I believe that it would be better to discuss comparisons between your study and other previous studies in the discussion, but if you decide to keep this table, please add references for each of the other locations.

Did the other locations have a eudominant species other than Ae. taeniorhynchus?

289: I suggest providing more info about the other papers; e.g. what kind of collection method did they use? Which seasons did they sample in?

**Conclusions**

-Are the conclusions supported by the data presented?

-Are the limitations of analysis clearly described?

-Do the authors discuss how these data can be helpful to advance our understanding of the topic under study?

-Is public health relevance addressed?

Reviewer #1: The study indicates that their findings are of use for health authorities which are not. Authors need to analyze mosquitoes of medical importance in detail.

Reviewer #2: Conclusions are very long and could be much more succinctly presented.

Reviewer #3: Some of the limitations require acknowledgement. For example, it should be acknowledged that most seasonal pattern studies are conducted over multiple years. Also, HLC is biased to mosquito species attracted to human hosts and therefore may limit the measured diversity beyond what is calculated as the percent coverage.

Otherwise, the conclusions are supported by the data presented, with the caveat that the statistical analysis should be updated to account for pseudo-replication. The authors do a relatively good job of putting their findings in the context of past studies, but the clarity of these parts of the discussion could be improved and additional details added (e.g. compare/contrast the collection methods and seasons for other YP mosquito biodiversity studies). The authors did a good job of addressing public health relevance in the discussion, although they should provide citations for the mosquito-pathogen associations throughout the MS (table and discussion). The authors go into a lot of detail in the discussion and it could be helpful to have a couple of summary sentences that tie all of the genus-level paragraphs together.

**Editorial and Data Presentation Modifications?**

Reviewer #1: The ms needs to be re-written and re-analyzed extensively.

Reviewer #2: The figures are generally visually appealing, but I would suggest that they be carefully examined to fix misspellings and that there be consistency between the labelling n the text and the figures. See specific comments below.

Reviewer #3: This paper could be improved by enhancing the quality of the writing to increase clarity. I provide some examples in the attached document, and hope that this will help to guide your edits to the paper. Some of the issues require changes to the order in which information is presented, especially in the introduction. Other issues require refining the writing to improve clarity and reduce redundancies. There are also grammatical errors throughout (for example, in the title, it should be “.. in a semi-conserved area in southern Mexico”).

The introduction section does not flow logically; I suggest reordering (e.g. start with an explanation of vector-borne diseases and zoonoses, then discuss mosquito diversity, and then the impact of the host community, and finally discuss the unique ecology and background info pertaining to the location.) This is just an example of a way that you could reorder it.

The writing throughout the manuscript needs to be tightened up; for example, lines 70-76 is very repetitive and could be edited to just one sentence. Also 85-87 is repetitive with 70-72. These are just a couple of examples, but there are similar types of repetitive writing throughout the paper. There are also typos (e.g. 116-118 should be “area” and “mangrove”).

**Summary and General Comments**

Reviewer #1: The ms needs to be focused substantially. Authors need to think that the journal is related to diseases which is a topic they talk very little in their work largely. They may always opt for a different journal and then remain with the same analysis.

Reviewer #2: This manuscript describes a sampling study conducted over a bit more than 1 year in a single area. The study uses the HLCs from 3 individual collectors to calculate a variety of diversity indices.

Notably, the biggest issue is the use of the word "anthropophagous" in the title of the study. The methods and results indicate that while HLCs were the predominant collection method, collections were also performed from dogs and inanimate objects which are not in alignment with anthropophagous. This needs to be addressed in revisions.

There are also calculated results described that are not described in the methods.

Specific comments are:

L2: Suggest title change to: "Diversity patterns of female mosquitoes in Southern Mexico." The collections were not performed just of anthrophilic mosquitoes.

L23-24: This sentence could be removed to increase clarity.

L32-33: This sentence is not needed in the Abstract. It is something that is fine to discuss but it is not something that represents a scientific result of this study.

L37-39: Suggest revision of the conclusion to be more impactful.

Text generally: Paragraph formatting is reversed in this review copy from normal paragraph formatting.

L56: Format reference to proper callout. Check throughout manuscript.

Study area: Did the three transects differ in vegetation or were they essentially the same?

L116 Should this be Study site and sampling transects?

L121: Suggest revision. Were the human settlements 546m from the perimeter of the sampling area?

Sampling section: Please specify which transect abbreviation used in figure 1 represents the mangrove, semideciduous, and modified semideciduous. Note the spelling of deciduous.

Also, is consent a requirement for HLC studies in Mexico? If so, it should be mentioned.

L159: If sampling was also sporadically conducted from dogs, then it is not quite accurate to characterize the effort as anthropophagic sampling.

L163-164: Carpenter & LaCasse reference callout?

L200-201: These collection methods are not mentioned in the sampling section above but should be. These also argue against characterization of the sampling as anthropophagic.

Figure 2 (Line 242): Calculation of Renyi diversity profile is not described in the methods. Also, please check spelling in the figure as well as reformat the panels so that they are the same size.

L247: Where are calculations that support this sentence and the others of this paragraph? I did not see them in a Table.

L252 & others: Generally, this manuscript would benefit from consistency in the naming between the text and figures. Here, they are NS, RS, DS but on the figure, they are norte, rainy, and dry.

L262: The text states, "the Hill numbers differed significantly" and then calls to Figure 4A. However, Fig 4A shows diversity, Shannon and simpson with 95% CIs. In the q=0 panel of 4A, the 95% CIs overlap so they are not significantly different. How can you thus conclude they are different? Similarly, most of the 95%CIs overlap in the panels of 4B.

L286: Definitely need to reference those previous studies here. Also needs explanation in the methods section if you are generating these diversity indices from previously published data.

Reviewer #3: Your manuscript reports an impressive amount of mosquito sampling and identification, which add substantively to the current literature on mosquito diversity on the Yucatan Peninsula. The main weakness is the clarity of the writing, which can be improved through a combination of simple editing and more involved reordering of information (especially in the introduction). The secondary weakness is the statistical analysis, which can be improved through using another analysis method that accounts for pseudo replication (e.g. GLMM) and providing more details about how the current calculations were conducted (e.g. R packages). I believe that your data is important to publish, and I encourage you to make the suggested changes to the writing and data analysis!

PLOS authors have the option to publish the peer review history of their article (what does this mean? ). If published, this will include your full peer review and any attached files.

**Do you want your identity to be public for this peer review?** For information about this choice, including consent withdrawal, please see our Privacy Policy .

Reviewer #1: **Yes: ** Alex Córdoba-Aguilar

Reviewer #2: **Yes: ** Alden Estep

Reviewer #3: No

**Figure resubmission:**

**Reproducibility:**



---

## [Decision Letter · Decision Letter 1]

Response to Reviewers
Revised Manuscript with Track Changes
Manuscript

Shaden Kamhawi

co-Editor-in-Chief

Paul Brindley

co-Editor-in-Chief

**Additional Editor Comments (if provided):**
**Journal Requirements:**
**Reviewers' comments:**

**Key Review Criteria Required for Acceptance?**

**Methods**

-Are the objectives of the study clearly articulated with a clear testable hypothesis stated?

-Is the study design appropriate to address the stated objectives?

-Is the population clearly described and appropriate for the hypothesis being tested?

-Is the sample size sufficient to ensure adequate power to address the hypothesis being tested?

-Were correct statistical analysis used to support conclusions?

-Are there concerns about ethical or regulatory requirements being met?

Reviewer #1: Statistical analyses were revised and now look fine.

Reviewer #2: See line comments below.

**Results**

-Does the analysis presented match the analysis plan?

-Are the results clearly and completely presented?

-Are the figures (Tables, Images) of sufficient quality for clarity?

Reviewer #1: Results are sound

Reviewer #2: Needs minor modifications. See line comments below.

**Conclusions**

-Are the conclusions supported by the data presented?

-Are the limitations of analysis clearly described?

-Do the authors discuss how these data can be helpful to advance our understanding of the topic under study?

-Is public health relevance addressed?

Reviewer #1: Conclusions are ok

Reviewer #2: See line comments below.

**Editorial and Data Presentation Modifications?**

Reviewer #1: (No Response)

Reviewer #2: Minor edits. See line comments below.

**Summary and General Comments**

Reviewer #1: Authors did a good job by using all reviewers' comments.

Reviewer #2: This manuscript examines mosquito diversity on Isla del Carmen over a period of one year. This revised manuscript accomplishes the goals noted in the abstract and needs only minor revisions. Line specific comments are noted below.

L23-24: A difficult sentence that could be revised.

L49: We examined host-seeking adult female mosquito diversity, abundance....

L57: Suggest dropping "despite it's dominance."

L114: Is this GPS location the centroid of the 10 hectares? Based on the figure, the sampling area is a bounded box so consider providing the GPS locations of the corners of the 10-hectare site.

L140-145: Clear statement of IRB approval for study and stated law for HLCs.

L164: Is there an accession/collection number for this deposit?

L341-347: This paragraph is not an experimental result so should not be in this section of the manuscript. It is quite appropriate as a part of the discussion.

L366: Suggest immature lifestages rather than Immature mosquito phases.

L406-408: I am glad to see this limitation stated. HLCs are going to be very biased toward species willing to feed on humans. Many Culex and Uranotaenia will not be sampled by HLCs.

I suggest that changing collection methods will not probably assist in understanding the drivers of activity but would help to better understand the full scope of mosquito diversity.

L415-417: Do you think the choice of HLC accounts for the relatively low percentages? What sampling methods were used in these other studies?

Sankey diagram: Species names in left column should be italicized. This should be a pretty easy mod to your R code.

PLOS authors have the option to publish the peer review history of their article (what does this mean? ). If published, this will include your full peer review and any attached files.

**Do you want your identity to be public for this peer review?** For information about this choice, including consent withdrawal, please see our Privacy Policy .

Reviewer #1: **Yes: ** Alex Córdoba-Aguilar

Reviewer #2: **Yes: ** Alden Estep

**Figure resubmission:****Reproducibility:** To enhance the reproducibility of your results, we recommend that authors of applicable studies deposit laboratory protocols in protocols.io, where a protocol can be assigned its own identifier (DOI) such that it can be cited independently in the future. Additionally, PLOS ONE offers an option to publish peer-reviewed clinical study protocols. Read more information on sharing protocols at https://plos.org/protocols?utm_medium=editorial-email&utm_source=authorletters&utm_campaign=protocols

---

## [Editor Report · Decision Letter 2]

Response to Reviewers
Revised Manuscript with Track Changes
Manuscript

co-Editor-in-Chief

Paul Brindley

co-Editor-in-Chief

**Additional Editor Comments :**

The AE has just a few suggestions for the Abstract and Author summary but recommended acceptance. Please make her suggested changes and upload and it will be accepted.

**Comments to the Authors:****Please note that the comments are uploaded as an attachment.****Figure resubmission:****Reproducibility:** To enhance the reproducibility of your results, we recommend that authors of applicable studies deposit laboratory protocols in protocols.io, where a protocol can be assigned its own identifier (DOI) such that it can be cited independently in the future. Additionally, PLOS ONE offers an option to publish peer-reviewed clinical study protocols. Read more information on sharing protocols at https://plos.org/protocols?utm_medium=editorial-email&utm_source=authorletters&utm_campaign=protocols

---

## [Editor Report · Decision Letter 3]

Dear Dr Canales Delgadillo,

We are pleased to inform you that your manuscript 'Abundance and divesity of host-seeking adult female mosquitoes in a coastal ecosystem in southern Mexico.' has been provisionally accepted for publication in PLOS Neglected Tropical Diseases.

Best regards,

Gisselle Vasquez

Guest Editor

Amy Morrison

Section Editor

Shaden Kamhawi

co-Editor-in-Chief

Paul Brindley

co-Editor-in-Chief

Thank you for making the recommended changes to the abstract and author summary sections of your manuscript. Your work provides unique and valuable mosquito diversity and abundance data for Isla del Carmen, in the southern Gulf of Mexico, having recorded 26 mosquito species including multiple vectors (over 80% species identified) in a 15-month period (September 2019-december 2020). Furthermore, the study identifies a season (norte) that promotes high mosquito activity. This research article provides valuable insights into mosquito ecology in coastal areas of southern Mexico and the implications for public health.

---

## [Editor Report · Acceptance letter]

Dear Dr Canales Delgadillo,

We are delighted to inform you that your manuscript, "Abundance and diversity of host-seeking adult female mosquitoes in a coastal ecosystem in southern Mexico," has been formally accepted for publication in PLOS Neglected Tropical Diseases.

Best regards,

Shaden Kamhawi

co-Editor-in-Chief

Paul Brindley

co-Editor-in-Chief
